**Cite this article:** Çimşir E, Şahin MD and Akdoğan R (2024) Unveiling the relationships between eco-anxiety, psychological symptoms and anthropocentric narcissism: The psychometric properties of the Turkish version of the Hogg eco-anxiety scale. *Cambridge Prisms: Global Mental Health*, **11**, e26, 1–9

anxiety; environmental health; global mental health; psychometric evaluations; society

**Corresponding author:**
Elif Çimşir;
Email: elifcimsir@anadolu.edu.tr

# Unveiling the relationships between eco-anxiety, psychological symptoms and anthropocentric narcissism: The psychometric properties of the Turkish version of the Hogg eco-anxiety scale

Elif Çimşir[1] 📷, Murat Doğan Şahin[2] 📷 and Ramazan Akdoğan[1] 📷

[1]Department of Guidance & Counseling Faculty of Education, Anadolu University, Eskisehir, Turkey and [2]Department of Measurement and Evaluation Faculty of Education, Anadolu University, Eskisehir, Turkey

## Abstract

The increasing number of losses and damages caused by the climate crisis has rendered the psychometric assessment of the climate crisis more important than ever, specifically in developing countries, such as Turkey. The aim of this study was to examine the psychometric properties of the Turkish version of the Hogg Eco-Anxiety Scale (HEAS-13), using exploratory structural equation modeling (ESEM) on the cross-sectional data collected from 445 adults (286 females and 159 males; Mage = 29.76, range 18–65). The results supported the four-factor solution of the original version in the Turkish sample. Further analysis confirmed the invariance of the HEAS-13 across genders. The results demonstrated significant correlations of the HEAS-13 subscales with the Brief Symptom Inventory (BSI) and the Anthropocentric Narcissism Scale (ANS), except for that between the behavioral symptoms subscale of the HEAS-13 and the ANS. Both the total and the subscale scores of the HEAS-13 were also found to be reliable, given the internal consistency and test–retest reliability values. The Turkish version of the HEAS-13 can expand the scientific understanding of eco-anxiety, which can help develop mental health services to mitigate the negative mental health impacts of the environmental crisis.

## Impact statement

The environmental crisis has a variety of negative consequences on mental health, including anxiety, obsessive thinking, distress, loss and grief. The psychometric assessment of eco-anxiety is thus more important than ever, particularly in developing countries such as Turkey, whose populations may be disproportionately impacted by the negative effects of the climate crisis. This study not only supports the psychometric validity and reliability of the Turkish version of the Hogg Eco-Anxiety Scale (HEAS-13), but it also shows that there are positive relationships between anxiety experienced in relation to the environmental crisis and mental health symptoms. Moreover, the study findings suggest that those who perceive human beings as being superior and more entitled than other species and the environment are less likely to experience eco-anxiety. Along with highlighting the need for increased attention to the negative mental health impacts of the environmental crisis, the results can help mitigate the environmental crisis.

## Introduction

The climate crisis, one of the major global issues of the present, has been endangering water and food sources, housing and health security, agriculture productivity and natural ecosystems all around the world (Sanson et al., 2019). It also has many effects on mental health, ranging from grief, loss and distress to emotional and behavioral issues and decreased mental health (Clayton et al., 2017; World Health Organization (WHO), 2023; Reyes et al., 2023). Recent research highlights climate anxiety (also referred to as eco-anxiety) as an important emotional response commonly experienced in response to the climate crisis (e.g. Clayton et al., 2014, 2017; Clayton, 2020; Clayton and Karazsia, 2020; Hogg et al., 2021; Reyes et al., 2023).

With the increasing attention to the mental health implications of the climate crisis, research regarding the anxiety experienced in relation to climate change and/or similar environmental issues has been accumulating (e.g. Dodds, 2021; Hickman et al., 2021; Stanley et al., 2021; Aruta and Guinto, 2022; Schwartz et al., 2023; Reyes et al., 2023). Although some researchers highlight the affective symptoms (e.g. Helm et al., 2018; Verplanken et al., 2020), others demonstrate that individuals experiencing this phenomenon also suffer from difficulties in the cognitive and physical/behavioral domains, such as obsessive thinking patterns, panic attacks, decreased

appetite and sleep disturbances (e.g. Clayton and Karazsia, 2020; Hickman, 2020; Hogg et al., 2021).

Suggesting that anxiety concerning environmental crises is not limited to climate change but should also include anxiety about various environmental catastrophes that may or may not be directly related to climate change, Hogg et al. (2021) investigated the construct of "eco-anxiety." Based on a mixed-method study, the researchers created a multifaceted eco-anxiety scale called the HEAS-13, which is the first and only validated scale of eco-anxiety that measures the anxiety concerning the global environmental crisis, rendering it different from the existing scales that focus solely on climate change anxiety (e.g. Clayton and Karazsia, 2020). The HEAS-13 is a 13-item self-report scale designed to capture four layers of eco-anxiety: affective symptoms, behavioral symptoms, anxiety about one's negative impact on the planet and rumination. The scale requires respondents to rate the frequency of experiencing the symptoms listed in the items when thinking about climate change and other environmental issues over the last 2 weeks, using a 4-point Likert scale (0 = *not at all*; 3 = *nearly every*).

Because the HEAS-13 is a psychometrically validated scale that captures the broad concept of eco-anxiety, researchers have already adapted it to other cultures, such as Italian (Innocenti et al., 2023; Rocchi et al., 2023) and German (Heinzel et al., 2023). In these adaptations, the original four-factorial model has been shown to be valid and reliable, except in the study conducted by Innocenti et al. (2023), which retained a one-factor structure. However, it is worth noting that, similar to much of the research on the mental health impacts of the climate crisis (e.g. Helm et al., 2018; Hogg et al., 2021; Stewart, 2021), the current adaptations of the HEAS-13 have been conducted primarily in Western countries. Considering that populations in less developed countries, such as Turkey, may have been more severely impacted by the climate crisis due to restricted access to resources that can mitigate its negative effects (American Psychiatric Association (APA), 2023), there is a pressing need for research from such countries. Consequently, we conducted this study with the aim of examining the reliability and validity of the HEAS-13 in a Turkish context.

This research was based on five hypotheses, the first being that the four-factor structure of the original English version of the HEAS-13 would be psychometrically supported in the Turkish sample similar to its other cultural adaptations (e.g. Heinzel et al., 2023; Rocchi et al., 2023). Anticipating that the construct of eco-anxiety remains consistent in structure and meaning across genders (see Putnick and Bornstein, 2016), our second hypothesis posited that the HEAS-13 would exhibit invariance between genders. The third hypothesis was that the HEAS-13 scores would be reliable. Our fourth and fifth hypotheses were produced to collect additional evidence for the construct validity of the HEAS-13 in the Turkish sample. To be specific, the fourth hypothesis was that the HEAS-13 would have significantly positive correlations with clinically relevant symptoms (i.e. psychological distress, anxiety, depression, somatization, obsession-compulsion, paranoid ideation, interpersonal sensitivity, phobic anxiety, hostility and psychoticism) as eco-anxiety has been shown to be moderately correlated with disruptive mental health outcomes (e.g. Hogg et al., 2021; Heinzel et al., 2023). Finally, our fifth hypothesis was that there would be a negative correlation between the HEAS-13 and the construct of anthropocentric narcissism, which, as the authors of this study, we define as "human beings' propensity to prioritize themselves over other living beings and the environment."

We anticipated a negative correlation between eco-anxiety and anthropocentric narcissism because individuals with high levels of anthropocentric narcissism are likely to perceive human beings as being superior and more entitled than other species, potentially leading to tendencies and actions that are less eco-friendly. Consequently, anthropocentric narcissists may exhibit lower levels of anxiety regarding the environmental crisis. As a result, this research will also help extend research aiming to understand important personality traits, such as anthropocentric narcissism, which potentially have a significant negative impact on the planet. This is specifically important considering that certain tendencies or "traits" are still poorly understood in terms of how they affect the ecosystem, despite being claimed as the primary causes of climate change (Milfont and Schultz, 2016; Evans, 2019; Fawzy et al., 2020; Logan and Prescott, 2022; Pitiruț et al., 2022).

## Method

### *Procedures and participants*

Following the university review board's approval, the data were collected from two sets of participant groups during the 2021–2022 spring semester. The first group of participants was the largest group (Sample 1, n = 385), which was recruited to test the validity of the HEAS-13. A number of participants (n = 169; 43.7%) in Sample 1 were recruited during classes taught by the first author of this study at a large university located in Turkey. The remaining participants in Sample 1 (216; 56.2%) were teachers with varying specialties (e.g. English, Math, Science, Music, Special Education, School Counselor, Arts) working in different schools located in Istanbul and Ankara, who were recruited during in-service training programs organized by the Ministry of National Education. All data were collected face-to-face, and participation was voluntary with only a small portion of participants (2–3%) refusing to participate. Of the 385 participants in Sample 1, with a mean age of 31.14 ranging from 18 to 65 (SD = 11.58), 236 (61.3%) were female, and 149 (38.7%) were male. The majority of the participants (n = 174; 45.2%) held a bachelor's degree, with the remainder comprising college students (n = 169; 43.9%) and graduate degree holders (n = 42; 10.9%).

A small sample of participants (Sample 2, n = 60) was recruited to explore the test–retest reliability of the HEAS-13 subscales. The participants in this group filled out the HEAS-13 twice at a two-week interval. The participants of this sample were junior (n = 26; 43.3%) and sophomore (n = 34; 56.7%) counseling students with the majority being female (n = 50; 83.3%).

### *Measures*

The data collection tool contained an informed consent form, a demographic information section with items regarding age, gender, occupation, or program of study, the Hogg Eco-Anxiety Scale (HEAS-13), the Anthropocentric Narcissism Scale (ANS) items and the Brief Symptom Inventory (BSI).

### *The Hogg eco-anxiety scale (HEAS-13)*

The HEAS-13 is a 13-item scale assessing four dimensions of the construct of eco-anxiety (i.e. affective symptoms, rumination, behavioral symptoms and anxiety about one's negative impact on the planet) on a four-category scale from "not at all" to "almost every day" (Hogg et al., 2021). Hogg et al. recommend that researchers obtain a mean score for each eco-anxiety subdimension, with greater scores demonstrating an increased average of incidence. In the context of the current study, the original version of

the HEAS-13 was independently translated into Turkish by two Turkish professors proficient in both Turkish and English, with expertise in mental health and well-being research. These two versions of the scale were then compared and the slight discrepancies were resolved. Following this, two graduate students who are fluent in both languages were contacted to compare the final version of the Turkish translation with the original version and to rate each item for both accuracy and clarity, on a scale from 1 (*not clear or accurate at all*) to 5 (*completely clear or accurate*). All items on the translated version received a score of five for both accuracy and clarity, confirming that the Turkish version of the HEAS-13 was consistent with the English version (see Supplementary Table S1, Supplementary Materials for both versions). This version of the scale was administered as the final Turkish version to a group of Turkish adults, along with some other questionnaires.

### Anthropocentric narcissism scale (ANS)

We developed the ANS (see Supplementary Table S2, Supplementary Materials), as part of this study, to assess the connection of the HEAS-13 scores to a negative environmental personality characteristic, anthropocentric narcissism, which we define as the inclination of certain individuals to prioritize human beings' interests and well-being over those of other living beings and the environment. Because anthropocentric narcissism is a new construct, we explored the items that we created through exploratory factor analysis (EFA) to discover their factor structure and internal reliability. This process resulted in a 7-item single-factor self-report questionnaire expecting individuals to indicate their level of agreement with its statements on a scale from 1 (*Strongly disagree*) to 7 (*Strongly agree*). The list of ANS items and item factor loadings (Supplementary Table S2), and the steps of the EFA (Supplementary Note) are presented in the Supplementary Materials.

### Brief symptom inventory (BSI)

The Brief Symptom Inventory (BSI; Derogatis and Melisaratos, 1983) involves 53 items developed to assess nine symptom dimensions: Somatization, Obsession-Compulsion, Interpersonal Sensitivity, Depression, Anxiety, Hostility, Phobic anxiety, Paranoid ideation, and Psychoticism. The scale also produces three global indices of distress (i.e. Global Severity Index, Positive Symptom Distress Index, and Positive Symptom Total) intended to measure the level of symptomatology, the intensity of symptoms and the number of reported symptoms. The BSI, used in various psychiatric and non-clinical settings (e.g. Pereda et al., 2007; Fernández et al., 2020), was adapted into Turkish by Şahin and Durak (1994) with robust validity and reliability, maintaining its original item number and factor structure. All nine of the symptom dimensions, as well as the global severity index (i.e. the most sensitive indicator of participants' distress level) of the Turkish version, were used in this study.

### Results

The data analyses included descriptive statistics and preliminary analyses, confirmation of the factor structure of the HEAS-13, measurement of gender invariance, as well as reliability and concurrent validity of the HEAS-13 scores.

### Descriptive statistics and preliminary analyses

The preliminary analysis involved assessments of missing values and the assumption of multivariate normality. Outlier detection

and multivariate normality tests were performed using the shiny application (Aybek, 2021), which identified 18 outliers and a violation of multivariate normality. As a result, the identified outliers were removed from the data set and the robust maximum likelihood (MLR) was used in all analyses using Mplus 8.0 (Muthén and Muthén, 1998–2017). Lastly, because the missing data (0.03%) were revealed to be missing completely at random (MCAR; $\chi2 = 2,742.08$; df = 2,715; $p = 0.35$), the missing values were also completed using the shiny application.

Item-level descriptive statistics (i.e. means, standard deviations, and skewness and kurtosis) of the HEAS-13 are shown in Supplementary Table S3 (see Supplementary Materials). Descriptive statistics as well as correlations between the HEAS-13 scores and the validity measures are also presented in Table 4. As seen in Table 4, the HEAS-13 subscales produced mostly moderate correlations with one another, except for that between behavioral symptoms and anxiety about the personal impact ($r = 0.28$, $p < 0.001$). The correlations mean that individuals experiencing higher affective symptoms are also likely to suffer from increased levels of behavioral symptoms ($r = 0.46$, $p < 0.001$), higher anxiety about their personal impact on the planet ($r = 0.48$, $p < 0.001$) and more time thinking about environmental problems ($r = 0.48$, $p < 0.001$). Moreover, both ruminating on environmental concerns ($r = 0.33$, $p < 0.001$) and being concerned about personal impact ($r = 0.28$, $p < 0.001$) translate into an increased number of behavioral symptoms.

As also seen in Table 4, it is noteworthy that none of the kurtosis and skewness values of the HEAS-13 fall outside the acceptable range (i.e. −3 to +3 for skewness and − 10 to +10 for kurtosis; Brown, 2006) for conducting structural equation modeling (SEM). Nevertheless, using a robust estimator (i.e. MLR) mitigates any potential issues, even if values were to fall outside the acceptable range. Additionally, a comparison of eco-anxiety scores with those of other countries is presented in Supplementary Table S4 (see Supplementary Materials). As indicated in the table, Turkish participants exhibit the highest mean scores on "Rumination" and "Behavioral Symptoms" compared to their counterparts in Italy, New Zealand and Australia.

Lastly, before proceeding with the main analyses of the study, we also checked if there were gender differences in the HEAS scores. The $t$-test results of affective symptoms, $t(383) = 2.68$, $p = 0.008$, show that females ($M = 0.98$, $SD = 0.55$) had significantly higher scores than males ($M = 0.82$, $SD = 0.63$). In contrast, males ($M = 1.18$, $SD = 0.55$) had higher rumination scores than females ($M = 0.98$, $SD = 0.72$), $t(383) = 2.81$, $p = 0.005$. There were no significant gender effects for behavioral symptoms, $t(383) = 0.08$, $p = 0.94$, with females ($M = 0.79$, $SD = 0.79$) and males ($M = 0.78$, $SD = 0.73$) scoring almost the same. Finally, females ($M = 1.23$, $SD = 0.64$) scored higher than males ($M = 1.05$, $SD = 0.67$) on anxiety about personal impact, $t(383) = 2.69$, $p = 0.007$.

### Confirmation of the factor structure of the HEAS-13

To collect evidence for the validity of the HEAS-13, we first tested the fit of the data to a one-dimensional structure using confirmatory factor analysis (CFA). Given that previous studies suggested that the HEAS-13 had four factors and that CFA is typically used when there is a clear understanding of the construct being measured, we then tested the four-factor structure also using CFA. However, other researchers argue that CFA is inadequate for psychological constructs since it is overly restrictive in that it only allows items to correlate with specific factors and not others (Marsh et al., 2016). Therefore, we also tested the four-factor structure

**Table 1.** Fit indices of the confirmatory factor analysis (CFA) and the exploratory structural equation modeling (ESEM) solutions of the HEAS-13

| Factor model | Fit index | Values |
|---|---|---|
| One factor (CFA) | RMSEA | 0.158 (0.147–0.169) |
| | CFI | 0.659 |
| | TLI | 0.591 |
| Correlated four factors (CFA) | RMSEA | 0.058 (0.045–0.071) |
| | CFI | 0.959 |
| | TLI | 0.945 |
| Correlated four factors (ESEM) | RMSEA | 0.032 (0.000–0.053) |
| | CFI | 0.993 |
| | TLI | 0.983 |

CFI, robust comparative fit index; RMSEA, robust root mean square error of approximation; TLI, robust Tucker–Lewis index.

using exploratory structural equation modeling (ESEM), which combines the explanatory structure of exploratory factor analysis (EFA) with the confirmatory perspective of CFA (Aparuhov and Muthen, 2009; Morin et al., 2020). We then compared these three models in terms of the comparative fit index (CFI), the

Tucker–Lewis index (TLI), and the root mean square error of approximation (RMSEA) to determine which model better fits the data. CFI and TLI values greater than 0.90 and 0.95, respectively, indicate adequate and excellent model fit, while RMSEA values less than 0.08 and 0.06, respectively, indicate adequate and excellent model fit (Hu and Bentler, 1999; Hooper et al., 2008).

Table 1 shows the values for the fit of the data to the one-dimensional CFA, four-dimensional CFA and four-dimensional ESEM models. The results show that the one-factor model produces poor fit indices (RMSEA = 0.158; CFI = 0.659; TLI = 0.591), whereas the four-factor CFA model has almost perfect indices (RMSEA = 0.058; CFI = 0.959; TLI = 0.945). The four-factor ESEM model, however, has a much better fit (RMSEA = 0.032; CFI = 0.983; TLI = 0.983). As a result, based on the fit indices of the four-factor ESEM model, the construct validity of the HEAS-13 can be suggested to be excellent, supporting our first hypothesis. The standardized parameter estimates for the CFA and ESEM solutions and the factor correlations can be seen in Table 2.

### Measurement of gender invariance

To investigate whether the same factorial structure applies to both genders, we conducted a gender invariance analysis using multiple-group exploratory structural equation modeling (mg-ESEM; Van De Schoot et al., 2015; Marsh et al., 2016) analysis on the four-factor

**Table 2.** Standardized factor loadings for the confirmatory factor analysis (CFA) and exploratory structural equation modeling (ESEM) solutions of the Hogg Eco-Anxiety Scale (HEAS-13)

| | | Correlated four factors | | | | | | | |
|---|---|---|---|---|---|---|---|---|---|
| | | CFA | | | | ESEM | | | |
| Items | One factor (CFA) | AS | RUM | BS | AAPI | AS | RUM | BS | AAPI |
| AS1 | 0.714** | 0.753** | – | – | – | **0.466**** | 0.075* | 0.014 | 0.003 |
| AS2 | 0.730** | 0.816** | – | – | – | **0.547**** | 0.022 | 0.089* | −0.047 |
| AS3 | 0.714** | 0.831** | – | – | – | **0.709**** | −0.066* | 0.006 | −0.047 |
| AS4 | 0.603** | 0.610** | – | – | – | **0.408**** | 0.011 | −0.090* | 0.144* |
| RUM1 | 0.637** | – | 0.829** | – | – | 0.031 | **0.604**** | −0.034 | 0.061 |
| RUM2 | 0.625** | – | 0.815** | – | – | 0.000 | **0.674**** | 0.033 | −0.071 |
| RUM3 | 0.620** | – | 0.776** | – | – | 0.003 | **0.558**** | 0.014 | 0.056 |
| BS1 | 0.513** | – | – | 0.719** | – | 0.079 | −0.001 | **0.570**** | 0.048 |
| BS2 | 0.528** | – | – | 0.775** | – | −0.042 | 0.093* | **0.556**** | 0.017 |
| BS3 | 0.494** | – | – | 0.794** | – | 0.072 | −0.074 | **0.698**** | 0.015 |
| AAPI1 | 0.563** | – | – | – | 0.670** | −0.023 | 0.026 | 0.192** | **0.495**** |
| AAPI2 | 0.536** | – | – | – | 0.761** | −0.020 | −0.035 | −0.004 | **0.628**** |
| AAPI3 | 0.530** | – | – | – | 0.696** | 0.104* | 0.089 | −0.106 | **0.506**** |
| Factor Correlations | | | | | | | | | |
| AS-RUM | | 0.560** | | | | 0.539** | | | |
| AS-BS | | 0.621** | | | | 0.536** | | | |
| AS-AAPI | | 0.566** | | | | 0.530** | | | |
| RUM-BS | | 0.357** | | | | 0.289** | | | |
| RUM-AAPI | | 0.671** | | | | 0.613** | | | |
| BS-AAPI | | 0.362** | | | | 0.250** | | | |

AS, affective symptoms; AAPI, anxiety about personal impact; BS, behavioral symptoms; RUM, rumination.
Bold values represent the loading to the specific factor.
*p < 0.05;
**p < 0.001.

**Table 3.** Results for the measurement invariance of the Hogg Eco-Anxiety Scale (HEAS-13) across gender

| Model | CFI | RMSEA (90% CI) | ΔCFI | ΔRMSEA |
|---|---|---|---|---|
| *Gender-Based ESEM Results* | | | | |
| Female | 0.999 | 0.009 (0.000–0.050) | | |
| Male | 0.968 | 0.073 (0.040–0.104) | | |
| *Multiple group ESEM models* | | | | |
| Model A: Configural invariance | 0.987 | 0.045 (0.016–0.066) | | |
| Model B: Metric invariance | 0.982 | 0.042 (0.020–0.060) | −0.005 | −0.003 |
| Model C: Scalar invariance | 0.981 | 0.041 (0.018–0.068) | −0.001 | −0.001 |

CFI, robust comparative fit index; ESEM, exploratory structural equation modeling; RMSEA, robust root mean square error of approximation.

ESEM model, given it was the best fitting model. Before proceeding to the steps of measurement invariance, the measurement model (i.e. ESEM) is initially applied separately for females and males. If the fit values obtained for both groups in this preliminary analysis are deemed acceptable, the process then advances to the stages of mg-ESEM.

Measurement invariance is tested using a four-stage model, the first of which is called configural invariance, in which all the parameters in both groups are freely estimated. If the indices of this model show at least an adequate fit, then the second stage, called metric invariance, is tested, by forcing equal estimation of factor loadings in both groups. If the metric invariance is also achieved, then scalar invariance is tested by constraining intercepts in both groups. Finally, if the scalar invariance is met, then the last step, called strict invariance, is investigated by constraining the error variances in both groups in addition to previous constraints. Some researchers claim that the last stage, strict invariance, is unnecessary when comparing latent variable means since error variances are no longer included in the latent variable and are therefore irrelevant (Vandenberg and Lance, 2000). Therefore, we omitted the last stage and tested the configural, metric and scalar invariances of the HEAS-13 in the present study.

As seen in Table 3, the fit values indicate an almost perfect fit for females (RMSEA<0.06; CFI > 0.95) and a good fit for males (RMSEA<0.08; CFI > 0.95). The mg-ESEM analyses indicated an almost perfect fit (RMSEA<0.06; CFI > 0.95) for all three of the configural, metric and scalar invariance. Additionally, the differences between metric versus configural (ΔRMSEA = -0.003 and ΔCFI = -0.005) and scalar versus metric (ΔRMSEA = -0.001 and ΔCFI = -0.001) were within 0.01 (Cheung and Rensvold, 2002). These results support our second hypothesis that the HEAS-13 would show measurement invariance across genders in the Turkish sample.

### Reliability analyses

The results supported our third hypothesis that the HEAS scores would be reliable. More specifically, the internal reliability value for the HEAS-13, as well as for its four subscales, was above the threshold (i.e. ≥0.70) with Cronbach's α = 0.87 for the total score, and internal reliability values changing between 0.74 and 0.85 for the subscales. Moreover, to support the stability of HEAS-13 scores over time ($n = 60$), we also calculated intraclass correlation coefficient (ICC) estimates for the total and the subscale scores, using a mean rating ($k = 2$), a 2-way mixed-effects model and absolute agreement (Koo and Li, 2016).

Contrary to Hogg et al. (2021), ruminating and experiencing anxiety about personal impact were less stable than affective and behavioral symptoms over time as shown by smaller ICCs: rumination: ICC =0.72, 95% CI = (0.53, 0.83); personal impact anxiety: ICC =0.74, 95% CI = (0.53, 0.84); affective symptoms: ICC =0.78, 95% CI = (0.63, 0.87); behavioral symptoms: ICC = 0.88, 95% CI (0.80, 0.93). Lastly, the total score of the HEAS-13 was also stable: ICC = 0.85, 95% CI = (0.75, 0.91). The ICC values indicate that the reliability of rumination and personal impact anxiety is moderate, while the scales for affective and behavioral symptoms, as well as the total HEAS-13 score, demonstrate good reliability. This evaluation is consistent with the criteria established by Koo and Li (2016), where ICC values below 0.5 are considered poor, those between 0.5 and 0.75 are deemed moderate, those between 0.75 and 0.9 are classified as good, and values above 0.9 are regarded as excellent.

### Concurrent validity

As seen in Table 4, the HEAS-13 exhibits moderate to large correlations (i.e. $r$ from 0.41 to 0.52; $p < 0.001$) with clinically relevant symptoms such as psychological distress, anxiety, depression, somatization, obsession-compulsion, paranoid ideation, interpersonal sensitivity, phobic anxiety, hostility and psychoticism, supporting our fourth hypothesis. Additionally, it shows a significant negative correlation with anthropocentric narcissism ($r = -0.19$, $p < 0.001$), which supports our fifth hypothesis. Regarding the relationships between the subscores of the HEAS-13 and BSI dimensions, the behavioral and affective symptoms, as well as anxiety about one's impact dimensions of the HEAS-13, show stronger correlations with clinically relevant symptoms (i.e. $r$ from 0.27 to 0.47, $p < 0.001$), while the rumination dimension shows significant but smaller correlations ($r$ from 0.13 to 0.22, $p < 0.01$) with the suggested symptoms.

Moreover, the correlations of the HEAS-13 subscales with anthropocentric narcissism were significantly negative ($r$ from −0.15 to −0.25, $p < 0.01$) except for the non-significant correlation between behavioral symptoms and anthropocentric narcissism. Consistent with our prediction, the most negatively correlated dimension of eco-anxiety to anthropocentric narcissism was anxiety about the personal impact ($r = -0.25$, $p < 0.001$). This suggests that individuals assuming more entitlement due to being a human species are significantly less concerned about their personal impact on the planet despite being no more or less likely to experience behavioral symptoms. In sum, the significantly negative correlation between the HEAS-13 and the ANS and the significantly positive correlation between the HEAS-13 and the BSI measures indicate that the HEAS-13 has good concurrent validity.

**Table 4.** Descriptive statistics and bivariate correlations between Hogg Eco-Anxiety Scale (HEAS-13) subscales, anthropocentric narcissism (AS), and brief symptom inventory (BSI) dimensions

| | | Dimensions of the HEAS-13 | | | | | | Descriptive statistics | | | | |
| | | AS | R | BS | AAPI | HEAS-13 | ANS | M | SD | Skewness | Kurtosis | Cronbach's α |
|---|---|---|---|---|---|---|---|---|---|---|---|---|
| Dimensions of the HEAS-13 | AS | – | | | | 0.82*** | −0.15** | 0.92 | 0.59 | 0.81 | 0.90 | 0.85 |
| | R | 0.48*** | – | | | 0.75*** | −0.19*** | 1.10 | 0.69 | 0.37 | −0.16 | 0.85 |
| | BS | 0.46*** | 0.33*** | – | | 0.69*** | 0.01 | 0.79 | 0.76 | 1.67 | 6.78 | 0.80 |
| | AAPI | 0.48*** | 0.53*** | 0.28*** | – | 0.74*** | −0.25*** | 1.16 | 0.66 | 0.16 | −0.26 | 0.74 |
| Dimensions of the BSI | GSI | 0.46*** | 0.22*** | 0.47*** | 0.39*** | 0.52*** | −0.06 | 61.54 | 38.71 | 0.64 | −0.17 | 0.96 |
| | S | 0.38*** | 0.20*** | 0.42*** | 0.28*** | 0.44*** | −0.02 | 6.14 | 5.53 | 1.01 | 0.61 | 0.86 |
| | OC | 0.40*** | 0.22*** | 0.39*** | 0.37*** | 0.46*** | −0.09 | 9.64 | 5.28 | 0.34 | −0.51 | 0.81 |
| | IS | 0.38*** | 0.17*** | 0.34*** | 0.34*** | 0.43*** | −0.08 | 4.89 | 3.91 | 0.77 | −0.20 | 0.81 |
| | D | 0.40*** | 0.20*** | 0.43*** | 0.37*** | 0.49*** | −0.08 | 8.23 | 5.62 | 0.57 | −0.35 | 0.86 |
| | A | 0.43*** | 0.22*** | 0.40*** | 0.35*** | 0.47*** | −0.08 | 6.49 | 5.67 | 2.56 | 18.08[a] | 0.80 |
| | H | 0.36*** | 0.13** | 0.37*** | 0.28*** | 0.39*** | 0.03 | 5.33 | 4.16 | 0.79 | 0.13 | 0.73 |
| | PA | 0.35*** | 0.17*** | 0.33*** | 0.27*** | 0.38*** | −0.05 | 4.29 | 3.97 | 1.03 | 0.40 | 0.78 |
| | PI | 0.37*** | 0.19*** | 0.33*** | 0.40*** | 0.43*** | −0.05 | 7.07 | 4.60 | 0.51 | −0.37 | 0.80 |
| | P | 0.41*** | 0.17*** | 0.42*** | 0.28*** | 0.44*** | −0.03 | 4.87 | 3.76 | 0.85 | 0.23 | 0.69 |
| | HEAS-13 | – | – | – | – | – | −0.19*** | 12.49 | 6.41 | 0.39 | 0.06 | 0.87 |
| | ANS | – | – | – | – | – | – | 14.87 | 7.65 | 0.74 | −0.15 | 0.82 |

A, anxiety; AAPI, anxiety about personal impact; AS, affective symptoms; BS, behavioral symptoms; D, depression; GSI, global severity index; H, hostility; IS, interpersonal sensitivity; OC, obsession–compulsion; P, psychoticism; PA, phobic anxiety; PA, paranoid ideation; R, rumination; S, somatization.
[a]Skewness or kurtosis outside of acceptable range.
** $p < 0.01$;
*** $p < 0.001$.

## Discussion

The current study aimed to assess the preliminary psychometric properties of the Turkish version of HEAS-13 (Hogg et al., 2021) in a sample of Turkish participants. The results of the ESEM supported that the Turkish version of the HEAS-13 has four subscales examining affective symptoms, rumination, behavioral symptoms and anxiety about one's negative impact on the planet. This finding aligns with the results from the study establishing the validity of the original instrument (Hogg et al., 2021). The intercorrelations between the HEAS-13 subscales were mostly medium, which implies that the presence of a specific eco-anxiety symptom does not necessarily correspond to the same degree of increase in another. These results are consistent with the findings of Hogg et al. (2021) as well as those of the researchers who adapted the scale to Italian (Rocchi et al., 2023) and German (Heinzel et al., 2023), in that different profiles may emerge as dimensions of eco-anxiety coexist.

Our findings also demonstrate that anxiety about one's personal impact and both affective and behavioral symptoms of eco-anxiety are moderately correlated with clinically relevant symptoms, supporting that the underlying components of eco-anxiety are similar to, yet also different from significant indicators of decreased mental health, such as depression, distress and anxiety (Hogg et al., 2021; Heinzel et al., 2023; Reyes et al., 2023). Furthermore, there were weak but significant correlations between the propensity to dwell on environmental issues and mental health symptoms (Cohen,

1988). This suggests that many individuals experiencing the ruminative component of eco-anxiety may not simultaneously suffer from significantly decreased mental health (Hogg et al., 2021), while also supporting previous research suggesting that while certain unfavorable environmental emotions may be strong indicators of decreased mental health, others may be weak yet still reliable (Ogunbode et al., 2023; Ojala et al., 2021; Stanley et al., 2021; Stewart, 2021). As a result, it is important to consider how varying aspects of eco-anxiety differentially relate to the concepts of health and wellness (Hogg et al., 2021).

It should be noted that this is the first study discussing the associations between the facets of eco-anxiety and an environmentally related personality trait, anthropocentric narcissism, which defines individuals' inclinations to prioritize human beings' interests and status over those of other living beings and the environment. Specifically, the results show that the HEAS-13 and its three subscales (affective symptoms, rumination and anxiety about personal impact) have inversely significant correlations with anthropocentric narcissism while none of the correlations between clinically relevant mental health symptoms and anthropocentric narcissism were significant. Aligning with our theoretical formulation, individuals with high levels of anthropocentric narcissism generally show less concern about their personal impact on the planet. They also exhibit lower levels of affective symptoms and a lower tendency to ruminate over environmental issues. However, they are neither more nor less likely to present clinically relevant

psychological symptoms. This supports the idea that anthropocentric narcissism is associated with ecologically relevant symptoms rather than mental health symptoms. These results also underscore the importance of further investigation into certain dispositions or "traits," such as narcissism, which may prove to be environmentally detrimental (e.g. Logan and Prescott, 2022).

This is also the first study examining and confirming the gender invariance of the HEAS-13. The results indicate that the latent factor structures of the HEAS-13 meet the requirement of stability, or "invariance" across genders, enabling valid comparisons between group means (see Van De Schoot et al., 2015). The validity findings, coupled with our results supporting the reliability of the HEAS-13, collectively establish its value as a robust measure for researchers and clinicians interested in assessing anxiety related to the environmental crisis and its correlates in Turkey. This is specifically important given that most of the studies regarding the mental health impacts of eco-anxiety have been conducted in developed countries, such as Australia, New Zealand, the United States, Italy and Germany (e.g. Hogg et al., 2021, 2023; Stewart, 2021; Heinzel et al., 2023; Rocchi et al., 2023) and that this trend excludes a significant percentage of the population living in developing (or under-developed) countries, such as Turkey, who may be disproportionately impacted by the environmental crisis (see APA, 2023; see also Rosa et al., 2021; Aruta and Guinto, 2022).

Our results also revealed that women's mean scores were significantly higher than men's in "Affective Symptoms" and "Anxiety about personal impact," aligning precisely with the results found recently in the Italian sample (Rocchi et al., 2023). This is consistent with Rocchi et al. (2023) that women may be more vulnerable to the symptoms of eco-anxiety. Furthermore, the comparison of mean scores among Turks, Italians, New Zealanders and Australians revealed that Turks had the highest mean scores on "Rumination" and "Behavioral Symptoms." Although this comparison was not subjected to a statistical significance test, we suggest that it can inspire further research exploring the manifestation of eco-anxiety across countries with diverse cultural and economic contexts.

Some limitations and further research directions apply to the results of this study. First, the set of studies depends on non-probabilistic samples of university students studying in the faculty of education and of practicing teachers, limiting the generalizability of our findings to individuals with less education and/or in other professions. Further research could thus examine the validity and generalizability of the HEAS-13 with various participant groups. Second, anthropocentric narcissism, which is an environmentally related personality trait that defines one's evaluations of human superiority over ecosystems and other species, is a new construct that was coined by the authors of this study. Although we explored the factor structure of the Anthropocentric Narcissism Scale (ANS), a 7-item self-report scale, as part of this study, future research should further validate the factor structure of the ANS, differentiate it from existing measures of environmentally related constructs (e.g. environmental beliefs, values, preferences, and attitudes; e.-g. Dunlap et al., 2000; Clayton, 2003; Mayer and Frantz, 2004; Schultz et al., 2005; Olivos and Aragonés, 2011; Steg et al., 2014; Parks-Leduc et al., 2015; Bouman et al., 2018; Wang et al., 2021) and conduct research on its determinants and outcomes. This may help prevent or appropriately intervene in intentions, actions and consumptions that are against sustainability and eco-friendliness. Also, because anthropocentric narcissism relates to one's cognitive evaluations regarding how supreme and exceptional human beings are in the natural world, it also bears some similarities to the construct of narcissism, which is characterized by feelings of self-importance, self-focus and grandiosity (Krizan and Herlache, 2018). As a result, future research may also benefit from contrasting and comparing the ANS with dark personality traits.

## Conclusion

This study has established the Turkish version of HEAS-13 as a valid and reliable measure of eco-anxiety, supporting a four-dimensional structure consistent with the original instrument. Medium intercorrelations between subscales indicated independent dimensions of eco-anxiety. Anxiety about personal impact and affective and behavioral symptoms demonstrated moderate correlations with clinically relevant symptoms, highlighting their unique yet interconnected nature. Higher anthropocentric narcissism levels were found to be associated with less eco-anxiety, underscoring the role of personality traits. Gender invariance of the Turkish version of the HEAS-13 indicated the validity of cross-gender comparisons. By enhancing our understanding of the links between eco-anxiety, personality traits and cultural factors, this study emphasizes the importance of inclusive research in addressing global environmental concerns.

**Open peer review.** To view the open peer review materials for this article, please visit http://doi.org/10.1017/gmh.2024.20.

**Supplementary material.** The supplementary material for this article can be found at https://doi.org/10.1017/gmh.2024.20.

**Data availability statement.** Data for this study are available at 10.6084/m9.figshare.23392442.

**Author contribution.** E.C. and R.A. prepared the theoretical background, conceptualized the study and collected data; M.D.Ş. analyzed the data and wrote the method section. E.C. prepared the first draft of the remaining sections. All authors commented on the drafts of the manuscript. All authors read and approved the final manuscript. We confirm that the order of the authors corresponds to the authors' relative contributions to the research effort reported in the manuscript.

**Financial support.** This research did not receive any financial support from funding agencies.

**Competing interest.** The authors declare none.

**Ethics statement.** Informed consent was obtained from all participants for inclusion in this study. The study was conducted in accordance with the Declaration of Helsinki, and the study was approved by the Ethics Committee of Anadolu University (#464306).

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
