## [Editor Report]

Revisor 1: The paper in general is moderate, with a lot of problems, which can preclude its publication in its current form. 

Introduction: The validated scale was not described sufficiently (i.e., its structure, dimensions, previous studies on psychometrics, etc.).

Hypotheses of the study are not clear. Please specify and number your hypotheses and justify them.

Please indicate clearly your aim of the study.

Methodology: The authors used the Anthropocentric Narcissism Scale (ANS), which was not used previously. They developed this scale for this study. Therefore, they checked validity with only the one established measure - BSI. But at the same time, they did not indicate that they used the Turkish version of the BSI. If they use only the Turkish translation, the paper can not be considered for publication, as convergent and divergent validity were checked with unestablished measures in a Turkish context.

When presenting measurement invariance, the authors should present fit indices in females and males separately, before checking configural, metric and scalar invariance.

Tables have a lot of abbreviations, which were not described in the notes. Therefore, it it unclear what the authors presented there (see for instance Table 2). Therefore, I can not assess the paper properly.

Second round of review is possible only when the authors address my comments adequately.

Abbreviations in the paper was used inattentively.

The methods used was described messily, insufficiently and illogically.

Descriptive statistics, as well as skewness and kurtosis values should be presented for all items/subscales of the validated scale.

The authors tested measurement invariance, but did not examine gender differences. They in general did not describe the descriptive statistics of the scale in order to understand the level of eco-anxiety. They did not compare (at least descriptively) these levels with levels in other countries.

Discussion: The authors should describe how the psychometrics of the scale is related to the theory of the construct measured. It is not sufficient to indicate the psychometric properties of the scale (moreover, the authors did not compare their results with the previous studies on the psychometrics of this scale).

Revisor 2: Overall, the article is well written and well presented. However, I suggest that the authors edit some minor typos and grammar check. (i.e. page 4 line 6 an “is” il lacking, line 46 there is an “those of” too many, page 6 line 18 “was separately translated...” may be paraphrased because the sense is not clear in english)

The results were well presented and statistical analyses were well conducted.

---

## [Editor Report]

Dear authors, we would like to congratulate you for presenting this article of great interest, as well as some recommendations that the authors have given us. We look forward to hearing from you 

Revisor 1: 

1. There are inconsistencies in using zeros in SM and in the paper.

2. Page 9: add p-value for correlations.

3. Page 13: Please use “r from XX to YY”, instead of using “~” or “–” when indicating a range. Page 12: add p-value for ICC.

4.Do not use “on the other hand” if you have no “on the one hand” before.

5. Table 2: “Standardized Parameter Estimates” mean Std. factor loadings?

6. Table 2: Please indicate whether the Factor Correlations are significant by providing p-values.

7. Table 4: Aplha for the GSI?

8. Please add a short conclusion. 

Revisor 2

I greatly appreciated the effort made by the authors in making important changes to the text which is now complete and scientifically very robust. As regards the Italian versions of The Hogg Eco Anxiety Scale and the studies conducted in Italy, I advise you to insert the citation and data regarding another Italian validation contained in I greatly appreciated the effort made by the authors in making important changes to the text which is now complete and scientifically very robust. As regards the Italian versions and the studies conducted in Italy, I suggest you to consider and report also data and citation of another Italian validation contained in an article published before that one of Rocchi et al. which you find here “Innocenti M, Perilli A, Santarelli G, Carluccio N, Zjalic D, Acquadro Maran D, Ciabini L, Cadeddu C. How Does Climate Change Worry Influence the Relationship between Climate Change Anxiety and Eco-Paralysis? A Moderation Study. Climate. 2023; 11(9):190. https://doi.org/10.3390/cli11090190”